# Wnt, Ptk7, and FGFRL expression gradients control trunk positional identity in planarian regeneration

**Rachel Lander[1], Christian P Petersen[1,2]\***

[1]Department of Molecular Biosciences, Northwestern University, Evanston, United States; [2]Robert Lurie Comprehensive Cancer Center, Northwestern University, Evanston, United States

**Abstract** Mechanisms enabling positional identity re-establishment are likely critical for tissue regeneration. Planarians use Wnt/beta-catenin signaling to polarize the termini of their anteroposterior axis, but little is known about how regeneration signaling restores regionalization along body or organ axes. We identify three genes expressed constitutively in overlapping body-wide transcriptional gradients that control trunk-tail positional identity in regeneration. *ptk7* encodes a trunk-expressed kinase-dead Wnt co-receptor, *wntP-2* encodes a posterior-expressed Wnt ligand, and *ndl-3* encodes an anterior-expressed homolog of conserved FGFRL/*nou-darake* decoy receptors. *ptk7* and *wntP-2* maintain and allow appropriate regeneration of trunk tissue position independently of canonical Wnt signaling and with suppression of *ndl-3* expression in the posterior. These results suggest that restoration of regional identity in regeneration involves the interpretation and re-establishment of axis-wide transcriptional gradients of signaling molecules.

**\*For correspondence:** christian-p-petersen@northwestern.edu

**Competing interests:** The authors declare that no competing interests exist.

## Introduction

Robust pattern control is a central but poorly understood feature of regenerative abilities (*Wolpert, 1969*; *French et al., 1976*). Animals cannot anticipate how a given injury will alter tissue composition, so regeneration likely depends critically on the re-establishment of missing tissue identity. Planarians use pluripotent stem cells to regenerate from nearly any amputation to restore a complete set of regionalized tissues, including cephalic ganglia in the anterior and a pharynx and mouth in the trunk, and are a model of positional restoration after amputation (*Reddien, 2011*; *Adler and Sánchez Alvarado, 2015*). Canonical Wnt signaling controls anterior-versus-posterior pole identity in planarian regeneration, with principal upstream determinants *wnt1* expressed at the posterior pole (*Petersen and Reddien, 2009*; *Gurley et al., 2010*) and the secreted Wnt inhibitor *notum* expressed at the anterior pole (*Petersen and Reddien, 2011*), both activated transcriptionally early after wounding. Inhibition of Wnt signaling components β-*catenin-1*, *wnt1*, *Evi/wntless*, *Dvl-1/2* and *teashirt* causes regeneration of ectopic heads (*Gurley et al., 2008*; *Iglesias et al., 2008*; *Petersen and Reddien, 2008*; *Petersen and Reddien, 2009*; *Owen et al., 2015*; *Reuter et al., 2015*); conversely, inhibition of *APC* or *notum* can cause regeneration of ectopic tails (*Gurley et al., 2008*; *Petersen and Reddien, 2011*). Other pathways participate in head or tail regeneration, with Hedgehog signaling influencing injury-induced *wnt1* expression (*Rink et al., 2009*), several transcription factors required for head formation (*prep, foxD, zic-1/zicA, pbx, egr-4*) (*Felix and Aboobaker, 2010*; *Blassberg et al., 2013*; *Chen et al., 2013*; *Fraguas et al., 2014*; *Scimone et al., 2014*; *Vásquez-Doorman and Petersen, 2014*; *Vogg et al., 2014*) and/or tail formation (*junli-1, pitx, pbx*) (*Blassberg et al., 2013*; *Chen et al., 2013*; *Currie and Pearson, 2013*; *Marz et al., 2013*; *Tejada-Romero et al., 2015*), and influenced by gap junction and calcium signaling (*Oviedo et al., 2010*;

**eLife digest** Some animals can regrow tissues that have been amputated. A group of flatworms called planarians are often used as a model to study the regeneration process because they are able to restore any lost tissue or even an entire animal from even tiny pieces of the body. For regeneration to be successful, it is critical to identify which tissues or regions of the body need to be replaced.

The planarian body is divided into three main parts: head, trunk and tail. Several genes involved in specifying what tissues regenerate are very active (or "highly expressed") in muscle cells in different regions of the planarian body. Some of the genes are involved in mechanisms that allow cells to communicate with each other, such as the Wnt and FGF signaling pathways. These genes could form a coordinated system to control regeneration, but their precise roles remain poorly understood.

Two groups of researchers have now independently identified genes that provide cells with information about their location in the flatworm body. Lander and Petersen identified three genes that are expressed in an overlapping manner along the body of uninjured animals. One of the genes – known as *ptk7* – is mainly produced in the trunk region, while the second gene (*wntP-2)* is produced from the tail and the third (*ndl-3)* is produced from the head region. The *wntP-2* and *ptk7* encode components of the Wnt signaling pathway, while *ndl-3* encodes a protein involved in FGF signaling.

Lander and Petersen used a technique called RNAi to lower the activity of the three genes individually or in pairs, and then examined whether this affected the ability of the worms to regenerate. Inhibition of any of the three genes resulted in an expansion of the trunk tissues into the tail region, indicating that the normal role for these genes is to stop cells adopting the trunk "identity".

Another study by Scimone, Cote et al. found that two separate sets of genes – including *wntP-2* and *ndl-3* – are needed to correctly position tissues in the head and trunk of planarians. Together these findings suggest that the Wnt and FGFRL pathways act in a body-wide system that co-ordinates where and which new tissues form during regeneration. A future challenge will be to decipher the complete network of genes that provides the positional information needed for regeneration.

*Zhang et al., 2011*). However, comparatively little is known about the restoration of positional information along the head-to-tail body axis through regeneration. Expression profiling and homology searching have identified a cohort of factors related to Wnt, Hox, and FGF signaling expressed regionally in domains along the anteroposterior (A-P) axis in planarians (*Cebrià et al., 2002*; *Petersen and Reddien, 2008*; *Reddien, 2011*; *Owen et al., 2015*; *Reuter et al., 2015*). These factors could in principle form a molecular coordinate system that re-specifies axis identity in regeneration and have been termed 'positional control genes' (PCGs) (*Witchley et al., 2013*). However, few phenotypes of positional displacement have been reported from perturbation of PCGs (*Cebrià et al., 2002*; *Kobayashi et al., 2007*), so it remains unclear how the majority of these genes participate in regeneration.

## Results

To uncover programs responsible for patterning along the body axis, we examined PCGs as defined by prior homology and expression profiling studies (*Petersen and Reddien, 2008*; *Reddien, 2011*; *Witchley et al., 2013*; *Owen et al., 2015*; *Reuter et al., 2015*) and found a unique expression pattern for a planarian homolog of *ptk7*, expressed in an animal-wide graded fashion maximally in the trunk, and also the CNS and pharynx (*Figure 1A*, *Figure 1—figure supplement 1*). Ptk7 proteins encode cell-surface Wnt co-receptors with a predicted intracellular pseudokinase domain that participate in noncanonical, β-catenin-independent Wnt signaling (*Lu et al., 2004*), and can either weakly suppress or activate canonical β-catenin-dependent Wnt signaling in a context-dependent manner (*Peradziryi et al., 2011*; *Puppo et al., 2011*; *Hayes et al., 2013*; *Bin-Nun et al., 2014*). Like other

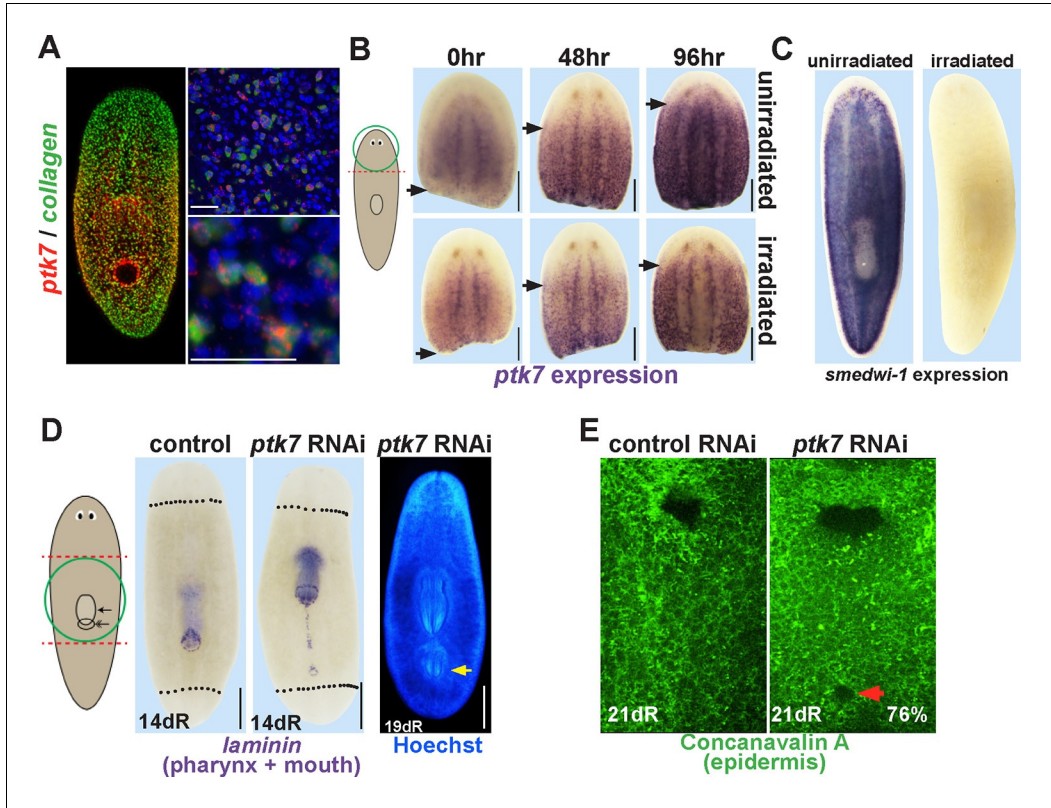

**Figure 1.** *ptk7* is a positional control gene that suppresses trunk identity in regenerating planarians. (**A**) Left panel, Double FISH to detect coexpression of *ptk7* within *collagen+* cells of the body-wall musculature in a trunk-centered gradient (116/125 *collagen+* cells were *ptk7+* and 113/125 *ptk7+* cells were *collagen+*, scored in ventral prepharyngeal subepidermal region). Right panels, higher magnification of *collagen+ptk7+* cells. (**B**, upper panels) Freshly amputated head fragments have *ptk7* expression in the CNS but minimal levels in subepidermal cells but by 48–96 hr expression appears at a region within the new anterior of the fragment (arrows, anterior extent of *ptk7* expression). (**B**, lower panels) Animals treated with lethal doses of gamma irradiation (6000 Rads) three days prior to amputation undergo a similar re-establishment of a *ptk7* expression domain along the A-P axis. Images represent at least 3/3 animals probed. (**C**) Irradiation controls showing elimination of *smedwi-1*-expressing neoblasts in animals from the same cohort as (**B**). (**D**) Animals were injected with control or *ptk7* dsRNA three times over three days, amputated to remove heads and tails, allowed to regenerate, fixed at 14 days and stained with a *laminin* riboprobe detecting both the mouth and pharynx (left panels, dotted line indicates amputation plane, red box shows enlarged region of *ptk7(RNAi)* animals) or stained with Hoechst dye to label nuclei and visualize the pharynx (right). *ptk7* RNAi caused formation of an ectopic posterior mouth in regenerating trunk fragments (28/35 animals), but not in regenerating head or tail fragments (35/35 animals each). (**D**, right) More rarely, *ptk7* inhibition caused formation of an ectopic posterior pharynx. (**E**) Control or *ptk7(RNAi)* animals stained with a fluorescent lectin Concanavalin A to visualize the epidermis and the pre-existing or ectopic mouth (red arrow). Bars, 25 (**A**), or 200 (**B**), or 400 microns (**D–E**).

The following figure supplements are available for figure 1:

**Figure supplement 1.** Sequence alignment of *Smed-ptk7*.

**Figure supplement 2.** RNAi enhancement screen identifies modulators of *ptk7* activity involved in trunk patterning.

described PCGs, planarian *ptk7* was expressed in *collagen+* cells of the body-wall musculature (*Witchley et al., 2013*) and aspects of its expression domain could become re-established after amputation even in irradiated animals lacking neoblasts and the ability to form new tissues (*Figure 1A–C*). In *ptk7(RNAi)* animals amputated to remove both head and tail, regeneration produced a normal head and tail (35/35) but caused formation of an ectopic posterior mouth at a high

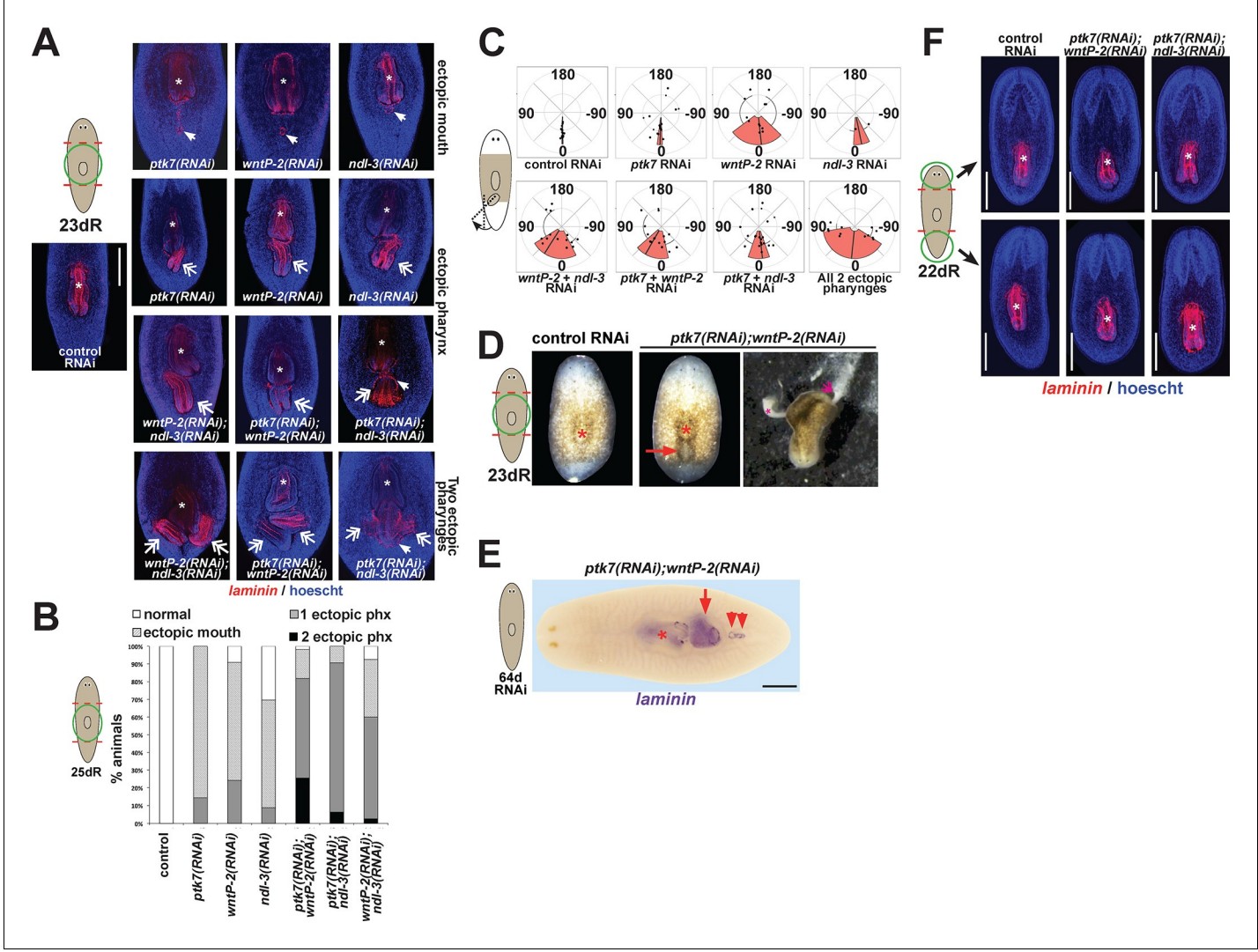

**Figure 2.** *ptk7* acts with *wntP-2* and *ndl-3* to suppress trunk identity in a context-dependent manner. (A) *ptk7*, *wntP-2*, and *ndl-3* dsRNAs were fed to animals individually or in pair-wise combinations prior to amputation to remove heads and tails, fixation 25 days later, staining with a *laminin* riboprobe and Hoechst dye. (B) Scoring information for pharynx and mouth duplication phenotypes. Animals were scored for presence of ectopic mouth (defined as a superficial circle of *laminin*+ cells which was always present posterior to the original mouth, arrow), and ectopic pharynx (defined as having pharynx morphology by *laminin*+ and Hoechst+ staining, double arrows) and its orientation with respect to the A-P body axis. Animals with an ectopic mouth but not a fully formed ectopic pharynx often had varying degrees of internal *laminin* expression suggestive of a growing pharynx primordium and were scored as having an ectopic mouth only. Co-inhibition of any pairwise combination of the three genes enhanced the penetrance and expressivity of the ectopic pharynx phenotypes. Note that combined pairwise inhibition of *ptk7*, *wntP-2* and *ndl-3* enhanced the trunk duplication phenotype and that dual inhibition of *ptk7* and *wntP-2* produced the strongest effects. (C) Analysis of ectopic pharynx orientation, measured at the proximal end of the ectopic pharynx. In many cases, the ectopic pharynx was oriented at an oblique angle with respect to the body axis, perhaps as a result of ectopic mouth placement nearby the original mouth, and ectopic pharynges were observed with fully inverted polarity. In all animals that formed 2 ectopic pharynges (derived from pairwise combinations of dsRNAs), both structures were oriented toward a common ectopic mouth located along the posterior midline. (D) Images of live animals showing the ectopic pharynx in *ptk7(RNAi);wntP-2(RNAi)* animals (red arrow) can be functional for feeding. (E) Prolonged inhibition of *ptk7* and *wntP-2* in uninjured animals for at least 36 days (64 days shown) caused formation of an ectopic pharynx (6/8 animals) and multiple posterior mouths (8/8 animals). (F) Inhibition of *ptk7* and *wntP-2* or *ptk7* and *ndl-3* caused head and tail fragments to regenerate only a single pharynx like control animals. Therefore, the effects of *ptk7*, *ndl-3* and *wntP-2* in body patterning are context dependent. Anterior, left. Bars, 300 (A,E) or 500 (F) microns.

The following figure supplement is available for figure 2:

**Figure supplement 1.** Verification of RNAi knockdown for *ptk7*, *wntP-2* and *ndl-3*.

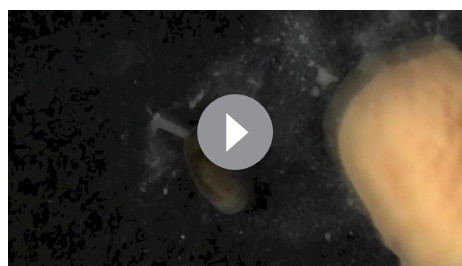

**Video 1.** Ectopic pharynges in *ptk7(RNAi);wntP-2(RNAi)* animals can be functional for feeding, related to *Figure 2*. Movie of a live *ptk7(RNAi);wntP-2(RNAi)* animal showing both the original and ectopic pharynges in a 21 day regenerating trunk fragment feeding on liver paste.

penetrance (83%, n=35) and, more rarely, formation of an ectopic posterior pharynx with broadly normal orientation with respect to the primary body axis (14%, n=35) (*Figure 1D–E*). Thus, *ptk7* limits trunk identity in planarian regeneration.

We next used RNAi and a phenotypic enhancement assay to identify other PCGs that participate with *ptk7* in trunk identity regulation (*Figure 1—figure supplement 2*). Co-inhibition of *ptk7* with either *wntP-2/wnt11-5/wnt4b* (hereafter referred to as *wntP-2)*, *ndl-3*, *Dvl-2* (*Almuedo-Castillo et al., 2011*), or *fzd-1/2/7* caused the strongest enhancement of the ectopic pharynx phenotype, resulting in 100% of animals affected, whereas co-inhibition with other PCGs affected this phenotype more weakly and below statistical significance under these conditions. *wntP-2* encodes a Wnt gene expressed in a graded fashion from the posterior

(*Figure 1—figure supplement 2*) and based on phylogenetic analyses has been proposed to be either a Wnt11 (*Gurley et al., 2010*) or Wnt4 (*Riddiford and Olson, 2011*) family member. Planarian *ndl-3* is expressed in a graded fashion from the anterior (*Rink et al., 2009*) (*Figure 1—figure supplement 2*), and encodes a member of the conserved FGFRL/*nou-darake* class of cell-surface molecules that possess an FGF-receptor-like extracellular domain but lacks a tyrosine kinase intracellular domain and are thus proposed to function as FGF signaling inhibitors (*Cebrià et al., 2002*; *Gerber et al., 2009*). *fzd-1/2/7* encodes a predicted Wnt receptor expressed broadly (*Figure 1—figure supplement 2*). Individual inhibition of *ptk7*, *wntP-2*, and *ndl-3* caused formation of ectopic posterior mouths (30/35 *ptk7(RNAi)* animals, 22/33 *wntP-2(RNAi)* animals, 14/23 *ndl-3(RNAi)* animals), and ectopic posterior pharynges (5/35 *ptk7(RNAi)* animals, 8/33 *wntP-2*(RNAi) animals, 2/23 *ndl-3 (RNAi)* animals) in amputated animals regenerating both their heads and tails compared to controls (0/46 animals) (*Figure 2A*). We verified knockdown of *ptk7*, *wntP-2*, and *ndl-3* using in situ hybridizations and qPCR (*Figure 2—figure supplement 1A–B*). Inhibition of any pairwise combinations of the three genes enhanced the trunk expansion phenotypes (*Figure 2A–B*) (animals with ectopic pharynges: 45/55 *wntP-2(RNAi);ptk7(RNAi)*, 29/32 *ptk7(RNAi);ndl-3(RNAi)*, 24/40 *wntP-2(RNAi);ndl-3 (RNAi)*), with double-RNAi animals also frequently forming two ectopic pharynges (*Figure 2A*) and co-inhibition of *ptk7* and *wntP-2* producing highest penetrance and expressivity of trunk duplication phenotypes. The orientation of ectopic pharynges was generally oblique, and sometimes inverted, but the majority in all conditions pointed toward the posterior rather than anterior direction (*Figure 2C*). Live animals with ectopic pharynges were observed during feeding, and the duplicated pharynx could obtain food (*Figure 2D*, *Video 1*), suggesting normal functionality of this organ. Taken together, *ptk7*, *wntP-2* and *ndl-3* form a core group of regionally expressed PCGs that jointly suppress trunk identity during posterior regeneration.

*wntP-2* and *ptk7* were expressed regionally in the absence of injury, so we examined whether they act only in regeneration or constitutively to regulate trunk regionalization. Prolonged co-inhibition of *wntP-2* and *ptk7* produced animals with an ectopic pharynx, indicating that these genes together restrict trunk identity homeostatically in the absence of injury (*Figure 2E*). Furthermore, the fact that such animals ultimately formed multiple ectopic mouths extending toward the posterior suggests that graded activities of *wntP-2* and/or *ptk7*, rather than their control of a binary switch in developmental outcomes, could pattern the tail and trunk regions.

We next examined whether trunk suppression mediated by *wntP-2*, *ptk7*, and *ndl-3* was operational in all regeneration contexts, as is the case for several characterized patterning genes in planarians. By contrast, under RNAi conditions that produced an 80–100% penetrant pharynx duplication in regenerating trunk fragments, head and tail fragments from *wntP-2(RNAi);ptk7(RNAi)* animals or *ndl-3(RNAi);ptk7(RNAi)* animals formed only a single *laminin+* pharynx as did control animals (*Figure 2F*). Eventually, after prolonged dsRNA feeding after regeneration, such animals could form ectopic pharynges (76 days of RNAi, n=3 of 12 animals examined), consistent with homeostatic

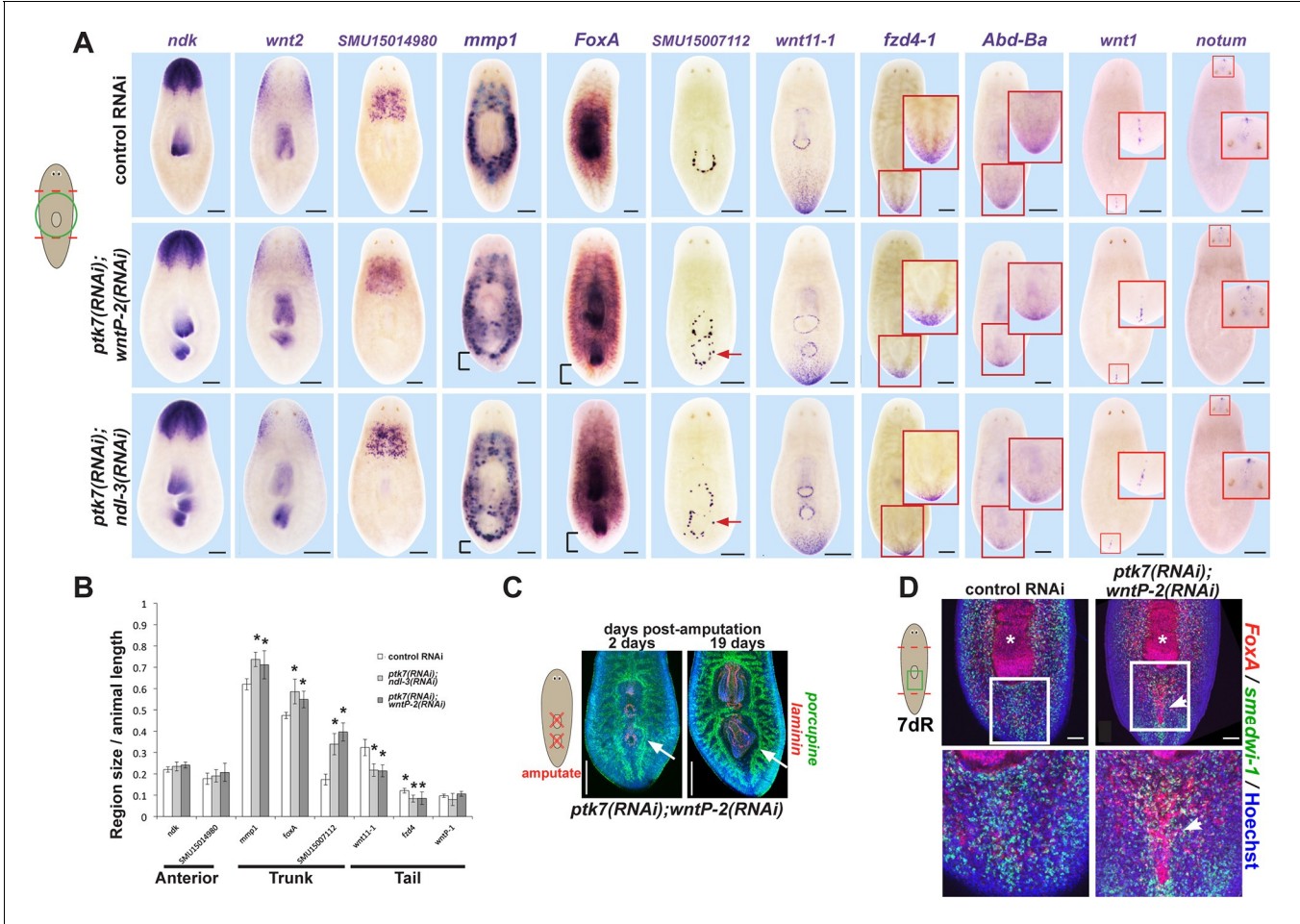

**Figure 3.** *ptk7*, *wntP-2* and *ndl-3* control tail-versus-trunk identity. (**A**) In situ hybridizations to detect A-P tissue regionalization in control and *ptk7 (RNAi);wntP-2(RNAi)* and *ptk7(RNAi);ndl-3(RNAi)* regenerating trunk fragments fixed 21 days after head and tail amputation, marking the anterior and head region (*ndk*, *wnt2*), prepharyngeal region (novel gene SMU15014980), trunk (novel gene SMU15007112, *mmp1*, *foxA*), posterior (*wnt11-1*, *fzd4-1*, *Abd-Ba*), and the anterior and posterior poles (*wnt1*, *notum*). All panels represent 100% of at least 6 animals stained. Arrow, ectopic trunk gene expression. Brackets, decrease in size of tail domain. (**B**) Quantitation of domain size changes from experiments described in (**A**), measured as length of domain normalized to body length. *ptk7(RNAi);wntP-2(RNAi)* and *ptk7(RNAi);ndl-3(RNAi)* regenerating trunk fragments had increased sizes of trunk domains marked by expression of *mmp1*, foxA and SMU15007112, and decreased sizes of tail domains marked by expression of *wnt11-1* and *fzd4-1* with little to no change to other domains. Asterisks, p<0.05 by 2-tailed t-test. (**C**) Both the pre-existing and ectopic pharynx in *wntP-2(RNAi);ptk7(RNAi)* animals regenerated (4/4 animals) after amputation with brief sodium azide treatment, using FISH to mark the gut (*porcupine*, green) and mouth and pharynx (*laminin*, red). Asterisk, pre-existing pharynx; arrows, ectopic pharynx, arrowhead, ectopic mouth. (**C**) *ptk7(RNAi);wntP-2(RNAi)* animals form ectopic FoxA+ cells by day 7 of regeneration. Bars, 100 (**D**), 200 (**A**), or 300 (**C**) microns.

The following figure supplement is available for figure 3:

**Figure supplement 1.** Additional histological analysis of *ptk7(RNAi);wntP-2(RNAi)* and animals.

functioning of the three genes. However, these results indicate that *ptk7*, *wntP-2* and *ndl-3* suppress trunk expansion in a context-dependent manner and suggest they may provide information about trunk absence or presence during regeneration.

We investigated the anatomy of *ptk7(RNAi);wntP-2(RNAi)* and *ptk7(RNAi);ndl-3(RNAi)* regenerating trunk fragments to determine the extent of the axis under control of the three genes. We first examined the influence of *ptk7*, *wntP-2*, or *ndl-3* inhibition on expression of PCGs and tissue-specific genes marking A-P axial identity (**Figure 3A–B**). Such *ptk7(RNAi);wntP-2(RNAi)* or *ptk7(RNAi);ndl-3 (RNAi)* regenerating animals had normal anterior pole and brain regions (marked by *notum*, *ndk*) and a normal pre-pharyngeal region anterior to the original pharynx (marked by *wnt2* and novel

gene *SMU15014980*), expansion of trunk-related peripharyngeal cells (expressing *mmp1*, *FoxA*, and *SMU15007112*), a reduced domain of PCGs expressed in the posterior (*wnt11-1*, *fzd-4-1* and *Abd-Ba*), and normal expression of *wnt1* at the posterior pole. We performed additional examinations of the brain (marked by *chat* and *cintillo*) and far posterior (marked by *wnt11-2*) in *ptk7(RNAi);wntP-2 (RNAi)* animals and found no apparent differences compared to control animals (*Figure 3—figure supplement 1A–B*). Thus, *ptk7*, *wntP-2*, and *ndl-3* normally promote anterior tail identity at the expense of the trunk and do not strongly affect head or tail formation. Both the pre-existing and ectopic pharynx were capable of regeneration after amputation by sodium azide treatment (*Adler et al., 2014*), suggesting that *wntP-2/ptk7* signaling acts in part to limit the size of the trunk region within the posterior rather than only functioning to position the anterior extent of newly made trunk tissue (*Figure 3C*). We next inhibited *ptk7* and *wntP-2* in a regenerating sexual strain of *S. mediterranea* that forms reproductive organs posterior to the pharynx upon attainment of appropriate size. Such animals formed both an ectopic pharynx and ectopic reproductive organs marked by *laminin* and *dmd-1* expression respectively (*Chong et al., 2013*), indicating *ptk7* and *wntP-2* regulate trunk regionalization beyond only control of pharynx and mouth formation (*Figure 3—figure supplement 1C*).

The pharynx is formed from *FoxA+* precursors derived from *smedwi-1+* neoblasts. Because inhibition of trunk identity genes produced an ectopic pharynx, we reasoned this structure likely arose from *FoxA+* progenitor cells. Indeed, *ptk7(RNAi);wntP-2(RNAi)* animals regenerating an ectopic pharynx produced an ectopic domain of *FoxA+* cells at a time (7 days of regeneration) prior to appearance of a fully formed ectopic pharynx (14 days of regeneration) (*Figure 3C*). Expression domains of *ptk7* (*Figure 1B*) and *wntP-2* (*Petersen and Reddien, 2009*; *Gurley et al., 2010*) can be altered by amputation independently of neoblasts, so we suggest these genes likely function to regulate axis organization upstream of controlling neoblast fates.

The unidirectional nature of these axis patterning phenotypes as expansion of trunk at the expense of tail identity prompted us to determine more precisely the nature of the regionalized expression of *ptk7*, *wntP-2* and *ndl-3* mRNAs. Visual inspection of colorimetric in situ hybridizations suggested that *ndl-3*, *ptk7*, and *wntP-2* are expressed in overlapping domains along the anteroposterior axis (*Figure 4A*). We verified this interpretation by quantifying the staining intensity of colorimetric in situ hybridizations along a lateral region running from head to tail (*Figure 4B*). *ndl-3* staining intensity was maximal in the anterior in a region of graded *ptk7* staining. *ptk7* staining was maximal in the trunk region in which *wntP-2* and *ndl-3* staining form opposing gradients. *wntP-2* staining was maximal in the posterior tail in a region of relatively less *ptk7* expression. The graded expression detected in this manner could arise from regional differences in the abundance of cells that uniquely express each factor or in regulation of cells that express combinations of the three genes. To test these models, we performed triple FISH to simultaneously detect expression of all three genes in eight domains along the head-tail axis, which broadly confirmed the regionalized expression behavior of anterior/pre-pharyngeal maximal *ndl-3* expression, trunk maximal *ptk7* expression, and tail maximal *wntP-2* expression (*Figure 4C*). We sought to verify this trend quantitatively and segmented the images to examine cells expressing any of the three genes (see Materials and Methods) then determined the mean FISH intensity for each gene per cell. This demonstrated that regions of maximal expression of *ndl-3*, *ptk7*, and *wntP-2* are comprised of cells with high expression of these genes (*Figure 4D*). We next examined pairwise comparisons of expression of each gene across the body axis (*Figure 4E*). This approach identified cells that co-express *wntP-2* and *ndl-3* (in particular in head-to trunk proximal regions R3-R5, *Figure 4E*), *ptk7* and *ndl-3* (head-to-trunk regions R2-R4), and *ptk7* and *wntP-2* (trunk-to-tail regions R5-R7). We explored whether this approach could identify the spatial distribution of discrete states of cells expressing all combinations of the three genes. We pooled the cell-based expression data across all regions to determine an approximate threshold to define higher versus lower expression for each gene, then assigned each measured cell into one of eight expression classes representing each combination of high versus low expression of each *ptk7*, *wntP-2*, and *ndl-3* and determined their distribution across the head-tail axis (*Figure 4—figure supplement1A–C*). This approach identified domains enriched for each expression state, finding a cohort with high *wntP-2* expression and low *ptk7* and *ndl-3* expression in the far posterior, a cohort co-expressing *ptk7* and *wntP-2* in the anterior tail and trunk regions, a cohort only expressing *ptk7* and not *wntP-2* or *ndl-3* centered in the trunk, cohorts of triple-positive cells and *ptk7+ndl-3+* cells in the pre-pharyngeal region, and cohort of *ndl-3+* cells in the anterior.

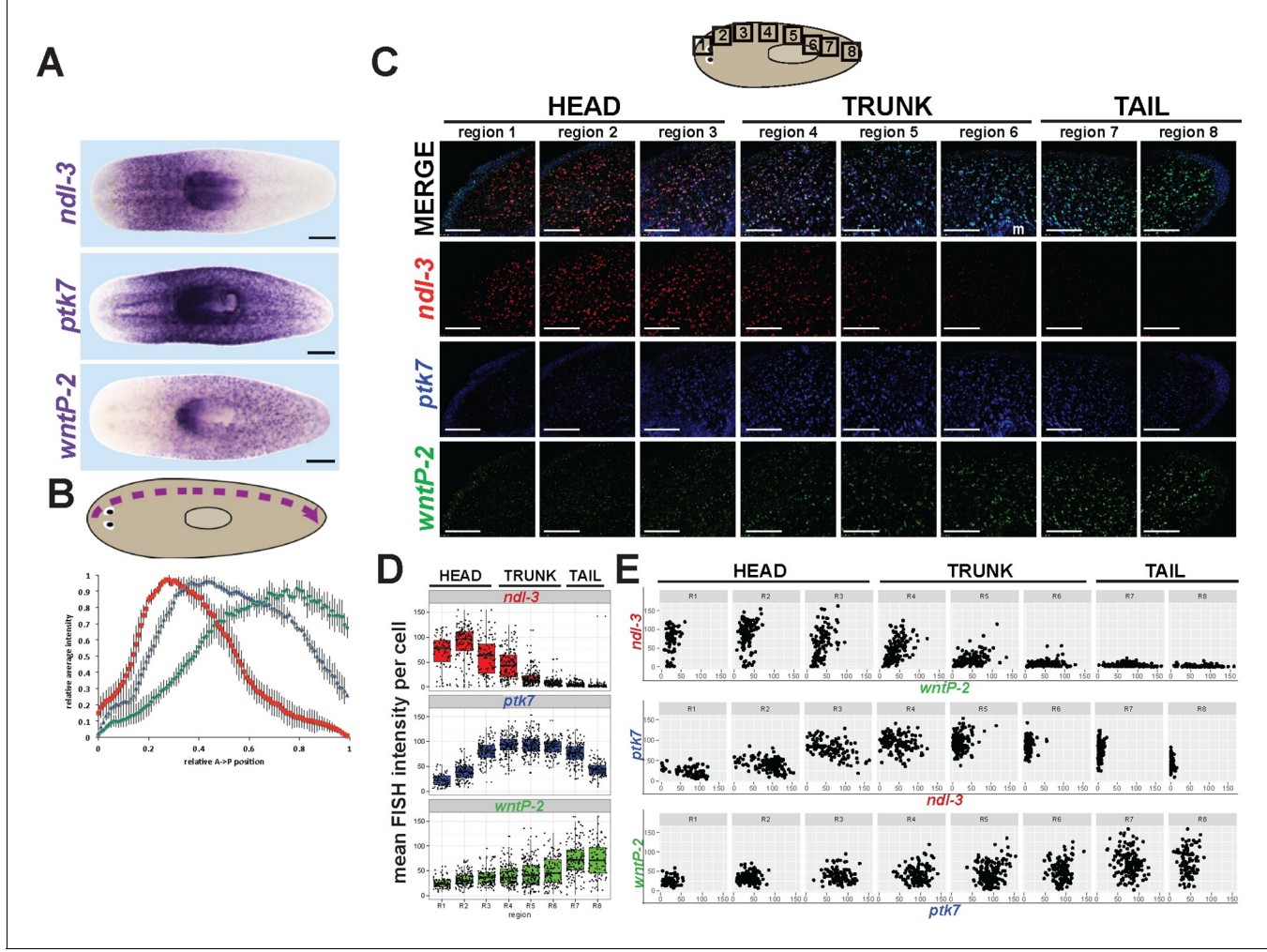

**Figure 4.** *ndl-3, ptk7*, and *wntP-2* are expressed in a graded fashion in domains along the anteroposterior axis. (A) In situ hybridizations showing body-wide graded expression of *ptk7* centered in the trunk, *wntP-2* expression in a gradient from the posterior and *ndl-3* expression in a graded fashion from the anterior. (B) Quantitation of colorimetric in situ hybridization staining across the body axis. 4–6 planarians stained as in (A) were imaged on a dissecting microscope, the images were inverted and then analyzed for position-specific staining intensity along a lateral domain depicted in the diagram (dotted line with arrow showing directionality). To compare animals of different lengths, position was normalized to length of this domain and signal intensity was normalized such that the minimum and maximum values across each animal were 0 and 1, respectively, and average intensity at each region was determined for animals stained with each probe treatment computed followed by obtaining average intensity, with bars showing standard deviations. (C) Triple FISH showing expression of *ndl-3* (red), *ptk7* (blue), and *wntP-2* (green) mRNA. Panels are maximum projections from a stack of seven 1-micron thick confocal images taken at 40x along the body axis at the regions represented in the cartoon, then adjusted for brightness and contrast uniformly for each channel across the image series. m, mouth. Bars, 100 microns. (D–E) Quantification of FISH signal intensity for cells identified in images shown in (C). 3-color images were segmented by merging all three channels to define a set of cells in each region with *wntP-2*, *ndl-3* and/or *ptk7* expression and this mask used to measure mean FISH signal intensity for each cell. (D) Scatter and box plots showing expression of *ndl-3* highest in the anterior, expression of *ptk7* highest in the trunk and tail, and highest *wntP-2* expression in the posterior. (E) Plots comparing pairwise FISH signal intensity between the indicated genes across eight body axis regions (R1-R8) as in (C). Note the existence of cells expressing both *ndl-3* and *wntP-2* (R3-R5), *ptk7* and *ndl-3* (R3-R4), and *wntP-2* and *ptk7* (R5-R7). Bars, 100 (C) or 200 (A) microns.

The following figure supplement is available for figure 4:

**Figure supplement 1.** Distribution of ptk7, wntP-2 and ndl-3 expression states across the body axis.

These results suggest that a complexity of PCG cell expression states populate regions of the body axis and that a region of high *wntP-2* and *ptk7* expression exists in the anterior tail at a location where these two genes act together to suppress trunk identity.

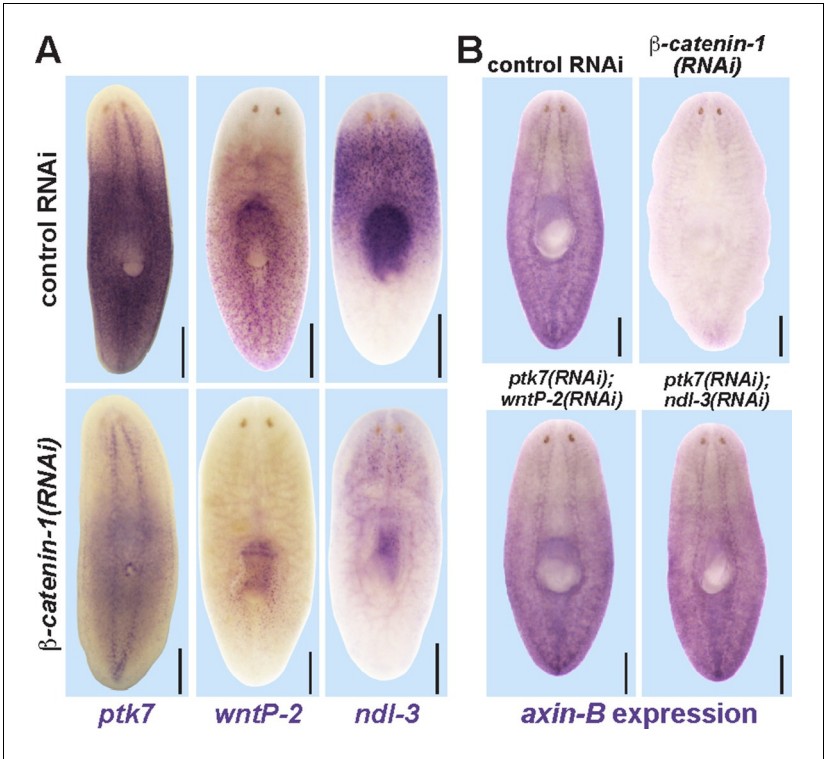

**Figure 5.** Trunk control genes likely signal independently of *β-catenin-1*. (**A**) In situ hybridizations show reduced expression of *ptk7* (11/11 animals), *wntP-2* (6/6 animals), and *ndl-3* (6/6 animals) after 8 days (*wntP-2, ndl-3*) or 19 days (*ptk7*) of *β-catenin-1* RNAi in uninjured animals. (**B**) In situ hybridizations showing reduction of *axin-B* expression after 11 days of *β-catenin-1* RNAi (14/14 animals) but not after inhibition of *wntP-2* and *ptk7* (14/14 animals) or *ndl-3* and *ptk7* (14/14 animals). Bars, 400 microns.

The following figure supplements are available for figure 5:

**Figure supplement 1.** Examining the effect of *APC* RNAi on expression of *ptk7*, *wntP-2*, and *ndl-3*.

**Figure supplement 2.** *fzd1/2/7* and *dvl-2* inhibition causes ectopic pharynx and mouth formation in the posterior.

**Figure supplement 3.** Testing planar cell polarity genes for involvement in trunk patterning.

**Figure supplement 4.** *ptk7*, *wntP-2,* and *ndl-3* inhibition do not influence *axin-B* expression and are not modified by *APC* inhibition.

We examined the involvement of canonical Wnt signaling on expression of these factors, as this pathway has multiple functions in patterning the primary body axis (*Petersen and Reddien, 2009*). *β-catenin-1* inhibition in uninjured animals severely reduced the expression of *ptk7*, *wntP-2* and *ndl-3* (*Figure 5A*), consistent with prior analyses of their expression requirements in regeneration (*Owen et al., 2015*; *Reuter et al., 2015*). We additionally inhibited *APC*, encoding an intracellular negative regulator of beta-catenin stability and examined regenerating animals for expression of the three trunk regulatory genes (*Figure 5—figure supplement 1*). Such animals regenerated anterior tails that expressed *wntP-2* throughout, that expressed *ptk7* strongly in a region near the amputation plane and away from the terminus, and that lacked *ndl-3* expression. These results suggest that beta-catenin upregulation can be sufficient for tail axis formation in conjunction with *wntP-2* and *ptk7* expression. Taken together, normal levels of beta-catenin signaling are important for the normal expression of *pkt7*, *wntP-2* and *ndl-3*.

We next examined candidates for signaling that could occur downstream of *wntP-2*, *ptk7*, and *ndl-3*. Ptk7 proteins can signal through several pathways, including as a coreceptor for Wnt/Frizzled

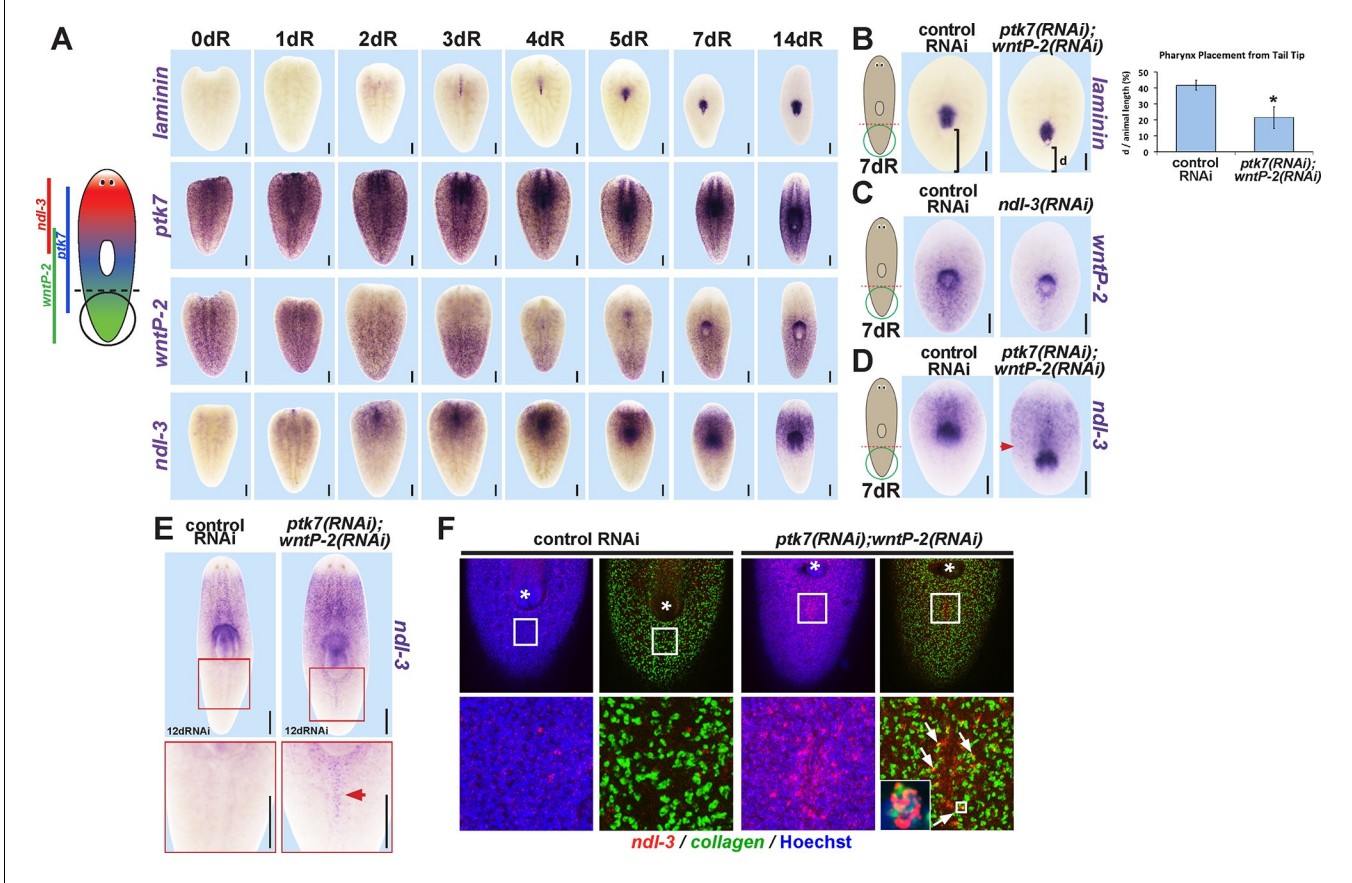

**Figure 6.** *ptk7* acts with *wntP-2* and *ndl-3* to specify trunk position in regeneration. (**A**) Cartoon shows regions of trunk control gene expression and in uninjured animals. In situ hybridizations of regenerating tail fragments showing that pharynx formation (marked by *laminin* expression) coincides with early reduction of *wntP-2* expression and increase in *ndl-3* expression. *ptk7* is expressed broadly and re-establishes a trunk-centered gradient by 7 days. All images represent at least 4/4 animals probed except *laminin* (d0) and *ptk7* (d0, d3 and d4) representing 3/3 animals probed. (**B**) *ptk7(RNAi);wntP-2 (RNAi)* animals form a single pharynx located too far posteriorly (7/8 animals, graph shows average distance between posterior pole and *laminin* expression domain normalized to animal length as in *Figure 6—figure supplement 1A–B*, error bars are standard deviations and asterisks shows p<0.05 by a 2-tailed t-test. (**C**) *wntP-2* expression is reduced in *ndl-3(RNAi)* regenerating tail fragments (10/14 animals). (**D-E**) *ndl-3* expression is expanded posteriorly in (**D**) *ptk7(RNAi);wntP-2(RNAi)* regenerating tail fragments by 7 days after amputation (4/5 animals) and in (**E**) intact animals after 12 or 17 days of RNAi (25/28 animals). (**F**) Control or *ptk7(RNAi);wntP-2(RNAi)* animals were stained for *ndl-3* and *collagen* expression after 17 days of RNAi and optical sections were imaged of the body-wall musculature in the region posterior to the pre-existing pharynx. Simultaneous inhibition of *ptk7* and *wntP-2* increased the frequency of *ndl-3+collagen+* cells versus total *ndl-3+* cells found in the tail region (41 of 63 cells in *ptk7(RNAi);wntP-2 (RNAi)* animals versus 7 of 32 cells scored in control animals, p<0.0001 by Fisher's exact test). Bars, 200 (**A–D**) or 400 (**E**) microns.

The following figure supplements are available for figure 6:

**Figure supplement 1.** *ptk7*, *wntP-2* and *ndl-3* participate in positioning the pharynx during tail fragment regeneration.

**Figure supplement 2.** Determining critical period for *ptk7/wntP-2* signaling in pharynx placement.

**Figure supplement 3.** Examining the influence of *ptk7*, *wntP-2* and *ndl-3* on each other's expression.

**Figure supplement 4.** Measurements of the influence of trunk control genes on *wntP-2* and *ndl-3* expression in tail fragment regeneration.

signaling (*Lhoumeau et al., 2011*). The results of our *ptk7* RNAi enhancement screen suggested a candidate molecular pathway in which *wntP-2* signals through *ptk7* and *fzd1/2/7* to suppress trunk identity. We verified that RNAi of *fzd1/2/7* alone resulted in ectopic mouth and pharynx formation similar to *ptk7* and/or *wntP-2* inhibition (*Figure 5—figure supplement 2A*). Similarly, we verified

that *Dvl-2* but not *Dvl-1* could interact with *ptk7* genetically, suggesting the involvement of *Dvl-2* in trunk suppression (*Figure 5—figure supplement 2B*). Ptk7 proteins can act in planar cell polarization so we tested for possible functional interactions between *ptk7* and components of the Planar Cell Polarity pathway *vangl1, vangl2, DAAM1* and *ROCK* but inhibition of these genes did not increase or decrease the occurrence of ectopic mouth or pharynx phenotypes (*Figure 5—figure supplement 3*). We additionally tested whether a downstream step in trunk suppression could be promotion or inhibition of beta-catenin activity. However, co-inhibition of *ptk7* and *wntP-2* or *ptk7* and *ndl-3* had no detectable effect on expression of beta-catenin-dependent transcripts *axin-B* and *teashirt* (*Figure 5B*, *Figure 5—figure supplement 4A–B*) (*Owen et al., 2015*; *Reuter et al., 2015*). *β-catenin-1(RNAi)* animals lose trunk regional identity and their pharynx while becoming completely anteriorized (*Iglesias et al., 2008*), so we tested for possible functional interactions between beta-catenin signaling and trunk control genes by inhibiting *ptk7/wntP-2* along with *APC*. We could not detect any enhancement or suppression of ectopic pharynx formation in that assay, further suggesting independence between *ptk7/wntP-2* and *APC/β-catenin-1* signaling (*Figure 5—figure supplement 4C*). These observations are consistent with the clear distinction between the *β-catenin-1* RNAi phenotype of ectopic head production (*Gurley et al., 2008*; *Iglesias et al., 2008*; *Petersen and Reddien, 2008*) as compared to the *ptk7(RNAi);wntP-2(RNAi)* and *ptk7(RNAi);ndl-3(RNAi)* phenotypes of ectopic trunk formation without affecting the identity of the anterior and posterior poles. Together, these results strongly suggest that trunk regionalization can be separable from pole identity and that *ptk7* and *wntP-2* likely do not operate exclusively through *β-catenin-1* to pattern the trunk and tail.

Regeneration can involve the re-definition of positional identity within pre-existing tissues, but the mechanisms controlling this process are unclear. We examined the expression and activities of *ptk7, wntP-2,* and *ndl-3* in forming a mouth and pharynx (expressing *laminin*) within the pre-existing tail tissue (*Figure 6B*). In amputated tail fragments, the mouth and pharynx were formed during the first 5 days of regeneration, with expression of *laminin* evident as early as 3 days. *wntP-2* was initially expressed throughout the amputated tail, but its expression restricted posteriorly starting around day 2, reaching a minimum around day 4, and re-establishing to intact proportions around day 7 (*Petersen and Reddien, 2009*; *Gurley et al., 2010*). *ndl-3* expression was initially absent in the tail fragments, emerged at day 1 in the far anterior then spread posterior to occupy the anterior half of the regenerating fragments by 4 days. *ptk7* expression remained broad in tail fragments through early times in regeneration and re-established a trunk-proximal domain evident by 7 days. Notably, the position of the newly regenerated pharynx correlated with a domain in which *wntP-2* expression was reduced and *ndl-3* was elevated.

We next examined the functions of *ptk7, wntP-2* and *ndl-3* in positional information control within regenerating tail fragments. In tail fragments, inhibition of *ptk7* caused a posterior shift to the location of the mouth and pharynx (*Figure 6A*, *Figure 6—figure supplement 1A*), an effect enhanced by co-inhibition of *wntP-2* or *ndl-3* though not caused by *wntP-2* or *ndl-3* inhibition alone (*Figure 6—figure supplement 1B*). Reduced doses of *wntP-2* and *ptk7* dsRNA resulted in intermediate placement phenotypes, and the distributions of placement phenotypes were not biphasic (*Figure 6—figure supplement 1C*), suggesting the activity of these genes could regulate pharynx position in a graded fashion rather than controlling a switch between two alternate organ locations. *ptk7(RNAi);wntP-2(RNAi)* tail fragments regenerated with posteriorly restricted expression of *fzd4-1* and *wnt11-1* and posteriorly expanded expression of *sFRP-2*, without strongly affecting the positioning of the brain (*ndk*) or pre-pharyngeal regions (*SMU15014980*) (*Figure 6—figure supplement 2D*). Together, these experiments suggest that *ptk7*, along with *wntP-2* and *ndl-3,* has a primary activity in controlling the positioning of trunk/tail tissues in regeneration.

To examine the relationship between the expression and function of *ptk7* and *wntP-2* signaling in regional identity re-establishment, we analyzed the time of emergence and critical time for pharynx placement phenotypes in regenerating tail fragments. RNAi of *ptk7* for three days prior to tail amputation resulted in a posterior shift in the location of the newly formed *laminin* expression domain detected as early as 3 days of regeneration (*Figure 6—figure supplement 2A*). These results indicate that *ptk7* acts before day 3 of regeneration. To determine a lower bound for the latest time of action for trunk control genes, we performed timed delivery of dsRNA via injections performed at successive two-day intervals during regeneration then examined the position of the *laminin* expression domain 7 days after amputation. Injections of *wntP-2* and *ptk7* dsRNA only prior to amputation,

or as late as day 1 and day 2, were capable of posteriorly shifting the *laminin* expression domain (*Figure 6—figure supplement 2B*). Taken together, these results suggest that *ptk7/wntP-2* signaling has functions in pharynx positioning after day 1 and before day 3, coinciding with the timing of *wntP-2*'s posterior restriction and *ndl-3*'s anterior expression. This occurs prior to the ultimate reestablishment of *ptk7*, *wntP-2* and *ndl-3* expression into finalized trunk, posterior, and anterior domains by 7–14 days, suggesting a separation between processes that control initial organ placement and ultimate proportion restoration (*Figure 6A*).

The coordinate regulation of *wntP-2* and *ndl-3* in regeneration led us to examine candidate transcriptional interactions among *wntP-2*, *ndl-3*, and *ptk7*. We did not detect strongly apparent changes to these expression domains after single inhibition of the other two genes by WISH, FISH or qPCR (*Figure 6—figure supplement 3A–C*). We tested for possible mutual expression requirements in tail fragments, reasoning this might provide a sensitized context in which the expression domains normally undergo regulation. *wntP-2* expression was reduced in *ndl-3(RNAi)* tail fragments, particularly along the lateral edges (*Figure 6C*), and posteriorly restricted in *ptk7(RNAi)* regenerating tail fragments (*Figure 6—figure supplement 4A*), suggesting that these genes can normally promote expression of *wntP-2* in regenerating tail fragments. By contrast, *ndl-3* expression was expanded posteriorly in *ptk7(RNAi)* and *ptk7(RNAi);wntP-2(RNAi)* tail fragments (*Figure 6D*, *Figure 6—figure supplement 4B*). We inhibited *ptk7* and *wntP-2* in uninjured animals to determine whether this effect was specific for regenerating tail fragments. Such animals expressed *ndl-3* ectopically posterior to

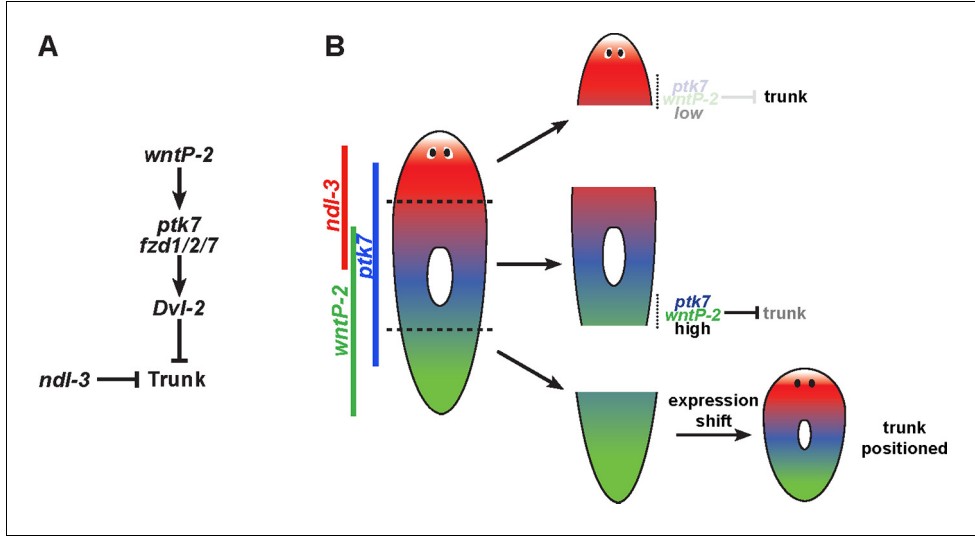

**Figure 7.** Model for *wntP-2*, *ptk7* and *ndl-3* in control of patterning. (**A**) A candidate molecular pathway of action in which *wntP-2* signals through *ptk7* and *fzd1/2/7* and *Dvl-2* to suppress trunk identity within the anterior tail region. The FGFRL *ndl-3* acts with the same sign as these components to suppress trunk regionalization and could act in a parallel pathway or modify the activity of the pathway through an unknown mechanism. *wntP-2* and *ptk7* can inhibit *ndl-3* expression in the posterior of regenerating tail fragments and intact animals and *ndl-3* promotes *wntP-2* expression in regenerating tail fragments, suggesting the potential for feedback regulation within this pathway (not shown). *β-catenin-1* signaling is required upstream for expression of *ptk7*, *wntP-2* and *ndl-3* (not shown). (**B**) Model relating expression of pathway components to patterning functions. The highest region of expression co-expression of *wntP-2* and *ptk7* occurs in the anterior tail and trunk at a location where these genes prevent trunk fates in animals undergoing tissue homeostatic maintenance and in regenerating trunk fragments that form new tail tissues. By contrast, regenerating head fragments lack abundant co-expression of *wntP-2* and *ptk7*, which we suggest is important for enabling the normal formation of trunk regional identity and regeneration of associated structures. Regenerating tail fragments would initially possess high levels of *wntP-2* and *ptk7* predicted to suppress trunk identity, but undergo a regeneration expression regulatory program that reduces *wntP-2* mRNA in this region, enabling trunk regeneration at a position that we suggest could be defined by a particular A-P location of *ptk7* and *wntP-2* activity present at an appropriate time in regeneration. According to this model, *wntP-2/ptk7* signaling provides information about the presence/absence of the trunk region used to control regeneration outcomes.

the pharynx at a time (by 12 days of homeostatic RNAi) prior to significant pharynx formation (*Figure 6E–F*, *Figure 6—figure supplement 3A–B*). Analysis of these ectopic *ndl-3+* cells by double-FISH revealed that the majority (41/63) co-expressed *collagen* and were located within the plane of the body-wall musculature (*Figure 6F*). These observations argue for a specific regulatory relationship in which *ptk7* and *wntP-2* together suppress expression of *ndl-3* in the posterior. Alternatively, the nascent pharynx could exert influence over the expression of *ndl-3* in a manner indirectly controlled by *ptk7* and *wntP-2*'s suppression of pharynx identity. Together these experiments indicate *ptk7*, *wntP-2* and *ndl-3* can directly or indirectly engage in regulatory interactions in homeostatic maintenance of tissue pattern and also in re-establishment of pattern in regeneration.

## Discussion

These experiments suggest a molecular model in which trunk identity and axis position are determined by low *wntP-2* activity signaling through the co-receptor *ptk7* and receptor *fzd1/2/7*, which together could provide competence for beta-catenin-independent outputs necessary for trunk and anterior tail patterning (*Figure 7A*). *ndl-3* is expressed in the trunk region yet acts with the same sign as *wntP-2* and *ptk7* to suppress posterior trunk expansion, either through a parallel or downstream process engaged in trunk suppression (*Figure 7A*) or perhaps due to its ability under some circumstances to promote robust expression of *wntP-2* (*Figure 6C*). In the tail region of uninjured animals and regenerating trunk fragments, high levels of *wntP-2/ptk7* activity prevent the acquisition of trunk identity and enable the formation of anterior tail tissue (*Figure 7B*). By contrast, head fragments initially have lower expression of *wntP-2* and *ptk7*, which could facilitate their formation of trunk tissue through regeneration. Amputated tail fragments would be expected to initially possess high levels of *wntP-2* and *ptk7* activity, but a regeneration expression regulatory program reduces *wntP-2* mRNA from the anterior to allow trunk regionalization to occur at an appropriate location. These models suggest that the expression status of *wntP-2* and *ptk7* could provide information about the presence or absence of pre-existing tissues used in determining regeneration outcomes. A previously proposed animal-wide gradient of *β-catenin-1* activity (*Reuter et al., 2015*) could set up the axis into distinct *ptk7*, *wntP-2* and *ndl-3* expression domains refined by *wntP-2* and *ptk7* repression of *ndl-3* expression. However, our experiments argue that a *β-catenin-1*-independent signaling output downstream of *wntP-2* and *ptk7* likely act to suppress trunk regional identity and thereby control placement of a trunk/tail boundary along the axis according to a gradation of their activities within the posterior. The identification of trunk expansion phenotypes independent of head/tail identity transformations suggests that whole-body regeneration involves a regulatory hierarchy of anterior/posterior pole formation followed by subsequent regional subdivision.

Our results thus establish a link between Wnt, Ptk7 and FGFRL proteins in regeneration patterning and axis formation. In mice and zebrafish, Ptk7 deletion causes defects in axis formation including a lack of convergent extension within the trunk and tail and a mispolarized auditory epithelium in mice, similar to disruption of core planar cell polarity components that signal independently of beta-catenin (*Lu et al., 2004*; *Hayes et al., 2013*). However, studies in *Drosophila*, zebrafish, *Xenopus* and mammalian tissue culture have found conflicting evidence that Ptk7 can also either promote (*Puppo et al., 2011*; *Bin-Nun et al., 2014*) or inhibit (*Peradziryi et al., 2011*; *Hayes et al., 2013*) beta-catenin-dependent signaling in a context-dependent manner. The regional identity defects we observe after *ptk7* RNAi in planarians are not obviously the result of defective planar cell orientations, are phenocopied by inhibition of a Wnt gene, and do not globally affect beta-catenin transcriptional targets or beta-catenin-dependent processes. We suggest that Ptk7 proteins can control tissue fate through an alternate mechanism, perhaps by coordinating the activities of cell cohorts within a field. Planarians and most other animals have expression of multiple Wnts in posterior domains, pointing to their ancient use in organizing the primary body axis (*Petersen and Reddien, 2009*). The use of Ptk7 proteins for trunk/tail regionalization could therefore have an ancient origin and allow posterior Wnts to produce distinct signaling outcomes for combinatorial pattern control.

We also find with *wntP-2* and *ndl-3* a second example in planarians of shared patterning regulation between Wnt and FGFRL factors. Whereas *wntP-2/wnt11-5/wnt4b* and *ndl-3* restrict the domain of the trunk, *wntA/wnt11-6/wnt4a* (*Kobayashi et al., 2007*) and *nou darake* (*Cebrià et al., 2002*) restrict the domain of the head and brain. Intriguingly, in both cases, the FGFRL genes have most prominent expression in the anterior but act with the same sign as the Wnt genes to promote more

posterior identity outside of the domain of their own expression. In mammals, FGFRL1 and Wnt4 are each required for formation of the metanephric kidney from intermediate mesoderm, suggesting this positive regulatory relationship is conserved (*Stark et al., 1994*; *Gerber et al., 2009*).

Pattern restoration in regeneration has been proposed to involve intercalation (*French et al., 1976*) or progressive specification of adjacent regions (*Roensch et al., 2013*) to restore positional values across a field disrupted by amputation. In planarians, asymmetric wound-induced expression of *notum* provides information about wound site directionality used to program anterior-versus-posterior pole fates (*Petersen and Reddien, 2011*). By contrast, we find genes encoding signaling molecules that are constitutively expressed in body-wide mRNA gradients and are used for control of positional information in regeneration. Graded expression of paracrine factors across fields of cells could enable patterning over the large length-scales necessary for adult regeneration. The interactions between this tissue-wide positional information present at the time of injury, combined with wound-induced directional cues, could account for robust pattern control in regeneration.

Note added in proof: while this manuscript was under review, Sureda-Gómez et al. reported that wnt11-5 RNAi causes a pharynx duplication phenotype (*Sureda-Gómez et al., 2015*).

## Materials and methods

### Planarian culture and irradiation treatments

Asexual and sexual strain *Schmidtea mediterranea* were maintained in 1x Montjuic salts between 18–20°C. Gamma irradiation (6000 Rads) was performed with a Cesium-137 source irradiator at least 24 hr prior to amputation to eliminate all dividing cells.

### Gene sequences

*Smed-protein tyrosine kinase-7* (*ptk7*) was identified through homology searches through BLAST on a planarian transcriptome database (Planmine, http://planmine.mpi-cbg.de) identifying a *Schmidtea mediterranea ptk7* homolog dd_Smed_v6_6999_0_1. *Smed-wntP-2* was described previously and cloned using primers 5'-TTAAATGTTCTAAGCCAAAACAACA-3' and 5'-AAAACTTTTATGATCAATC TGAATGC-3' (NCBI accession number: EU29663) (*Petersen and Reddien, 2009*). *Smed-ndl-3* was cloned using primers 5'-TTATTGACAGTAGGAACCAAAGCC-3' and 5'-ATCCTGAATCAAG TCAACGCCA-3' for dsRNA and riboprobe synthesis as described (*Rink et al., 2009*). Unless otherwise noted, riboprobes for a 4313-bp fragment of *ptk7* were made from a PCR product cloned by RT-PCR into pGEM-T-easy vector using the primers 5'-GTACTACCTGCCGAAAGTATACA-3', 5'-GCGCATATTCTATTGTGTAACGC-3'. In Figure S7, *ptk7* riboprobe was synthesized from a 2068-bp fragment using the primers 5'-CGACTGTTAGTTGGTTTATGGAC-3', 5'-ACTTGCCTTCTC TTTGAGCG-3'. SMU15014980 and SMU15007112 (*Robb et al., 2015*) (http://smedgd.stowers.org) expression patterns were identified by in situ hybridization screening from genes defined as BPKG22168 and BPKG1900 by prior RNA-seq studies (*Labbé et al., 2012*). SMU15014980 was cloned using the primers 5'-GGATGCTTTTGCATTTTGCT -3' and 5'-ATTGGCAAGAAAGCCATGAG -3'. SMU15007112 was cloned using the primers 5'-CCCCGTGTGGATATTTCAGT -3', 5'-AGCAAAA TCGGTTCTCCGTA -3'. For inhibition of *ptk7*, dsRNA was synthesized from a 1412-bp cDNA fragment cloned by RT-PCR into pGEM vector using the primers 5'-TGCTGGAAATAGTCTGTTGCAT-3', 5'-AAGATGGAACCCCAATAGCC-3'. Control dsRNA was generated from a 1527-bp fragment of *Photinus pyralis luciferase* from the pGL3-control vector (Promega, Fitchburg, WI USA) (primers 5'-TATCCGCTGGAAGATGGAAC-3', 5'-CGGTACTTCGTCCACAAACA- 3'). *wntP-2* (*Petersen and Reddien, 2009*), *ndl-3* (*Rink et al., 2009*), *laminin* (*Adler et al., 2014*), *porcupine* (*Rink et al., 2009*), *dmd-1, FoxA* (*Adler et al., 2014*), *smedwi-1, ndk, wnt1* (*Petersen and Reddien, 2009*), *collagen* (*Witchley et al., 2013*), *chat, mag-1, cintillo, β-catenin-1* (*Petersen and Reddien, 2008*), *APC* (*Gurley et al., 2008*), *wnt11-1* (*Petersen and Reddien, 2008*), *wnt11-2* (*Petersen and Reddien, 2008*), *wnt11-4/wntP-3* (*Petersen and Reddien, 2008*), *wnt2* (*Petersen and Reddien, 2008*), *wnt5* (*Gurley et al., 2010*), *fzd4-1* (*Petersen and Reddien, 2008*), *fzd4-2, fzd-1/2/7, Dvl-1* (*Gurley et al., 2008*), *Dvl-2* (*Gurley et al., 2008*), *axin-B, teashirt* (*Owen et al., 2015*), *notum* (*Petersen and Reddien, 2011*), *sfrp-1* (*Gurley et al., 2010*), *sfrp-2* (*Gurley et al., 2010*), *sfrp-3* (*Gurley et al., 2010*), *DAAM1* (*Beane et al., 2012*), *ROCK* (*Beane et al., 2012*), *vangl1, vangl2* (*Almuedo-Castillo et al.,*

*2011*; *Beane et al., 2012*), and *ndl-4* (*Rink et al., 2009*) riboprobes and dsRNAs were described previously.

## Fixations and stainings

Animals were fixed and stained as previously described (*Pearson et al., 2009*; *King and Newmark, 2013*). In brief, animals were killed in 5% N-acetyl-cysteine in 1xPBS for 5 min and then fixed in formaldehyde for 20 min at room temperature. Subsequently, animals were bleached overnight (~16 hr) in 6% hydrogen peroxide in methanol on a light box. Digoxigenin- or fluorescein-labeled riboprobes were synthesized as described previously (*Pearson et al., 2009*). Colorimetric (NBT/BCIP) or fluorescence in situ hybridizations were performed as previously described (*Pearson et al., 2009*; *King and Newmark, 2013*). For FISH, blocking solution was MABT with 10% horse serum and 10% western blot blocking reagent (Roche) (*King and Newmark, 2013*). Riboprobes were detected using anti-Dig-HRP (1:2000), anti-FL-HRP (1:2000), anti-DNP (1:2000), or anti-Dig-AP (1:4000) antibodies. Hoechst 33,342 (Invitrogen) was used at 1:500 as counterstain. Images of colorimetric staining were acquired using a Leica M210F scope with a Leica DFC295 camera and adjusted for brightness and contrast. Fluorescence imaging was performed on a Leica DM5500B compound microscope with Optigrid structured illumination system for optical sectioning or a Leica SPE confocal microscope. Images are maximum projections of a z-series with adjustments to brightness and contrast using Photoshop or ImageJ. Concanavalin A conjugated to AlexaFluor 488 (Invitrogen) was used to stain the epidermis as described (*Zayas et al., 2010*).

## Image analysis

For *Figure 4B*, animals stained with NBT/BCIP for detection of *ptk7*, *wntP-2*, and *ndl-3* were visualized with a Leica M210F dissecting microscope. Images were inverted and an intensity profile obtained from a segmented line drawn along a lateral region running from the anterior to posterior of the animal with a width approximately 1/6 of the animal width using ImageJ ('plot profile'). To make comparisons of intensity plots across animals of different sizes, position along the segmented line was normalized to its length. Background, taken as the minimum intensity along this profile, was subtracted, and the resulting values were normalized to the maximum intensity along this profile. This axis was divided into 100 equal sized bins to allow comparing intensity measurements across animals of different sizes, and the average normalized intensity was determined for each bin. These plots were compared for 4–6 animals stained with each riboprobe to generate an average and standard deviation of position- and global intensity-normalized in situ hybridization signal and plotted in *Figure 4B*.

In *Figure 4C*, Triple FISH was used to simultaneously detect expression of ptk7, wntP-2, and ndl-3 by obtaining maximum projections of 1-micron thick confocal images taken at 40x at 8 regions. The resulting three-color images were processed in ImageJ to convert to a merged grayscale image used for automated threshholding ('Auto Threshhold' Li's Minimum Cross Entropy method [*Li, 1998*]) and segmentation ('Analyze particles') using empirically optimized parameters (size=0.06–0.40 inch2 in the image, corresponding to 24–166 microns2 in the sample, and circularity 0–1.00). The resulting segmented areas were manually inspected to verify that the majority surrounded individual cells. Mean intensity per segmented area was determined for each channel and plotted using R (ggplot2) in *Figure 4D* and *Figure 4E*. In *Figure 4—figure supplement 1A*, density histograms were plotted to determine a cutoff for estimating high and low expression of each gene within each analyzed cell and used to classify cells as expression positive or negative. This assigned each cell to one of 8 classes plotted as a fraction of total cells in each region in a histogram in *Figure 4—figure supplement 1B* and as a scatterplot in *Figure 4—figure supplement 1C*. Similar trends were observed when threshold was taken to be any of 10 different threshold values between intensities of 30 and 75.

## RNAi protocols

RNAi treatments were performed either by dsRNA injection or feeding. dsRNA was synthesized as described previously (*Petersen and Reddien, 2011*). Unless otherwise noted, RNAi by injection was performed using a Drummond microinjector to deliver 5 x 32 nL dsRNA on three consecutive days, followed by transverse amputations and regeneration for the indicated number of days (*Figures 1D,*

*E*, *Figure 1—figure supplement 2*, *Figure 2D,E*, *Figure 3A,D*, *Figure 3—figure supplement 1A,B*, *Figures 5—figure supplement 1–3*, *Figure 6A–D*, *Figure 6—figure supplement 1,2A,4*). For experiments involving RNAi by feeding, animals were given a mixture of liver paste and in vitro transcribed dsRNA, as described (*Rouhana et al., 2013*). In brief, animals were fed every 2–3 days for either 1 week (*Figure 2—figure supplement 1*, *Figure 5*, *Figure 5—figure supplement 4*, *Figure 6E,F*) or 2 weeks (*Figure 2A*, *Figure 6—figure supplement 3*) and were maintained homeostatically or allowed to regenerate for the indicated number of days prior to fixation. For long-term RNAi treatment in the absence of injury, animals were fed a mixture of liver paste and dsRNA every 2–3 days for 2 weeks, followed by one dsRNA feeding every subsequent week (*Figure 2F*). For RNAi treatment in sexual *S. mediterranea*, animals were fed dsRNA every 2–3 days for one week and amputated transversely to create trunk fragments containing the pharynx and reproductive structures including the copulatory apparatus then fed dsRNA once per week for 62 days (*Figure 3—figure supplement 1C*). For timed dsRNA delivery experiments (*Figure 6—figure supplement 2B*), RNAi-treated animals on days -2, -1, 0 were injected with dsRNA on three consecutive days, amputated transversely to create tail fragments on day 0, and animals fixed on day 7 of regeneration. For subsequent timed delivery of dsRNA, animals were amputated transversely to create tail fragments on day 0, followed by dsRNA injection for two consecutive days as indicated, and fixed on day 7 of regeneration.

## Chemical amputation of pharynx in RNAi-treated animals

Animals were fed an equal mixture of *ptk7* and *wntP-2* dsRNA every 2–3 days for one week, amputated to remove heads and tails and allowed to regenerate for 20 days to produce an ectopic pharynx, then both the pre-existing and ectopic pharynges were removed by treatment with 100 mM sodium azide as described previously (*Adler et al., 2014*). Animals were then fixed either 2 days or 19 days later and stained with *laminin* and *porcupine* riboprobes to assess regeneration of the ectopic pharynx (*Figure 3C*).

## qPCR

Total RNA was extracted by mechanical homogenization in Trizol (Invitrogen) from three RNAi-treated intact animals, and purified in three biological replicates for each treatment. RNA samples were DNased-treated (TURBO DNase, Ambion) and cDNA was synthesized using SuperScript III reverse transcriptase (Invitrogen). qPCR was performed using SYBR Green PCR Master Mix (Applied Biosystems). *axin-B* mRNA was detected using the primers 5'-TTCCAGTTCAGGTCACATCG-3' and 5'-CATTGACACCTTCCGAACCT-3', *ubiquilin* mRNA was detected using 5'-AAATTCGCCTGCCTGTTGGG-3' and 5'-CCGGTGGCATTAATCCATCTGT-3', *clathrin* was detected using 5'-GACTGCGGGCTTCTATTGAG-3' and 5'-GCGGCAATTCTTCTGAACTC-3', *wntP-2* was detected using 5'-TGCTAAATCAACACCAGAATCAGCT-3' and 5'- CACATCCACAATTACTATGCACCCC-3', *ndl-3* was detected using 5'- CTCCCACAATTTATGAGTGCGGT-3' and 5'- TCTTGGGCCAATTTTGAGTTTTGATCTA-3', and *ptk7* was detected using 5'- GATCAAATCCCAAATCCAGTTC-3' and 5'-GGGTTTCTGGGAGTTTATATCGTA-3'. Relative RNA abundance was calculated using the delta-Ct method after verification of primer amplification efficiency. p-values were computed from a 2-tailed t-test.

## Acknowledgements

We thank members of the Petersen lab for helpful discussions and E Hill for the *ndl-3* plasmid.

## Additional information

### Funding

| Funder | Grant reference number | Author |
| --- | --- | --- |
| National Cancer Institute | Training grant T32-CA080621-09 | Rachel Lander |
| National Institutes of Health | NRSA F32GM108395-01A1 | Rachel Lander |

| Ellison Medical Foundation | New Scholar in Aging Research Award AG-NS-0835-11 | Christian Petersen |
| American Cancer Society | Institutional Research Grant ACS-IRG 93-037-15 | Christian Petersen |
| National Institutes of Health | NIH Director's New Innovator Award 1DP2DE024365-01 | Christian Petersen |

The funders had no role in study design, data collection and interpretation, or the decision to submit the work for publication.

## Author contributions

RL, Acquisition of data, Analysis and interpretation of data, Drafting or revising the article; CPP, Conception and design, Acquisition of data, Analysis and interpretation of data, Drafting or revising the article

## Author ORCIDs

Christian P Petersen, http://orcid.org/0000-0001-7552-6865

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
