## [Decision Letter]

Thank you for submitting your work entitled "Wnt, Ptk7, and FGFRL expression gradients control trunk positional identity in planarian regeneration" for consideration by *eLife*. Your article has been reviewed by three peer reviewers, and the evaluation has been overseen by Alejandro Sánchez Alvarado as Reviewing Editor and Fiona Watt as the Senior Editor. One of the three reviewers has agreed to reveal her identity: Yukiko Yamashita.

The reviewers have discussed the reviews with one another and the Reviewing Editor has drafted this decision to help you prepare a revised submission.

Summary:

In the manuscript by Lander and Petersen, the authors show that the freshwater planarian uses the kinase-dead WNT co-receptor, *ptk7*, to specify a novel trunk domain during axial regeneration. It has been a long-standing question as to how signaling pathways can act over long distances such as those across several millimeters of the planarian anterior-posterior axis. The work here begins to answer this question by identifying patterning molecules for large body domains. The authors further show that a previously known WNT molecule and an FGFRL molecule also play a role to pattern large domains around the central *ptk7* region. When knocked down by RNAi, these molecules and other potential positional controlling genes (PCGs) together with Ptk7 identified 4 genes (*WntP2, ndl-3, Dvl-2* and *fzd-1/2/7*) that potentiate the phenotype which appear to form a co-regulatory circuit that particularly affects the positioning of the pharynx and mouth. The authors nicely show that Ptk7 together with *ndl-3* or *WntP-2* are not required for extreme anterior or posterior pole identity, and that these genes function independently of β-catenin signaling. Finally, the authors show that pharynx positioning is dependent on this system during tail regeneration.

Essential revisions:

Overall, the claims made in this paper are well supported and the data is novel and interesting. However, a more detail-oriented rewriting of this manuscript should be considered and the following major concerns should be addressed before moving forward with publication.

1) One point of confusion is that RNAi of *ptk-7*+*WntP-2* and *ptk-7+ndl-3* are used interchangeably. Given the significant difference in penetrance shown in Figure 1—figure supplement 6, I am not convinced that these two contexts always generate the same phenotype.

2) Figure 3 nicely shows that *ptk7, WntP-2* and *ndl-3* expression are affected by β-catenin RNAi. However, because the patterning of more posterior tissues appears to be controlled by *WntP2, ndl-3* and *ptk7*, I think it is important to also show how expression of these genes are affected in posteriorized animals.

3) The claim (in Figure 4) that *ndl-3* expression is controlled by feedback is not well supported. Another possibility is that the pharynx position dictates the posterior boundary of *ndl-3* expression, rather than feedback between Wnt signaling and FGFRL signaling.

4) Do head or tail fragments eventually form ectopic mouths/pharynges? If the axial patterning is altered, I would expect to see this during regeneration too.

5) The authors observe RNA "gradients" with their 3 main genes of interest, but what evidence do they have that this is a protein gradient?

6) If *ptk7* specifies "trunk" as the authors strongly state, then why no anterior expansion when trunk identity is removed? Brain should be assayed. Trunk expansion and not loss of trunk should give ectopic pharynxes, yet the authors report the opposite. Perhaps it is that *ptk7* restricts *FoxA*+ stem cells, and when knocked down, that population expands and induces another pharynx.

7) Dvl and *fz1/2/7* are stated to also enhance the *ptk7* phenotype, but were dropped from analysis. Why?

8) Almost no stains are quantified. RNAi efficacy is not measured by qRT-PCR or RNAseq as is standard in the planarian field. This would greatly help in making such statements as in “These results suggest that β-catenin upregulation can be sufficient for tail axis formation in conjunction with *wntP-2* and *ptk7* expression. Taken together, normal levels of β-catenin signaling are important for the normal expression of *pkt7*, wntP-2 and ndl-3”.

9). In vertebrates, *ptk7* has been associated with WNT, semaphoring/plexin, and VEGF signaling, in addition to its known roles in PCP. None of this was tested here to bolster mechanism. Perhaps PCP defects can explain aspects of the phenotype. The authors state that the ectopic pharynxes are properly oriented, but many examples shown do not look that way.

10) More cell type markers are needed for tissues in different regions to conclude certain axial areas are not expanding/contracting.

11) Overlap in expression of *ptk7, wntP-2* and *ndl-3* should be quantified. How is the overlap changing in different RNAi conditions? Why would these genes overlap in the cells that express them, should it be a binary expression state of muscle?

12) Are the ectopic pharynxes functional?

---

## [Author Response]

*Essential revisions: Overall, the claims made in this paper are well supported and the data is novel and interesting. However, a more detail-oriented rewriting of this manuscript should be considered and the following major concerns should be addressed before moving forward with publication. 1) One point of confusion is that RNAi of ptk-7+WntP-2 and ptk-7+ndl-3 are used interchangeably. Given the significant difference in penetrance shown in Figure 1—figure supplement 6, I am not convinced that these two contexts always generate the same phenotype.*

We have altered the text throughout to clarify the conclusions we draw from these two experimental contexts. We also performed additional histological analysis of *ptk7(RNAi);wntP-2(RNAi)* animals and also *ptk7(RNAi);ndl3(RNAi)* animals. Such animals have a similar frequency of forming ectopic pharynges (80% vs 90% of animals, Figure 2), a similar distribution of ectopic pharynx orientations (Figure 2), and a similar expression of axial markers (Figure 3) that show enlargement of trunk (*mmp1, FoxA*, SMU15007112) and reduction of anterior tail (*wnt11-1, fzd4-1, Abd-Ba*). These genes may act in other distinct processes in the animals or influence trunk/tail regionalization in subtly distinct ways, but based on these analyses we could not find any consistent differences in the *ptk7(RNAi);wntP-2(RNAi) versus ptk7(RNAi);ndl3(RNAi)* phenotypes. We re-examined our scoring for formation of 1- versus 2- ectopic pharynges in these two treatments and found variability in this ratio across experiments, which we believe to be due to differences in feeding status and size of the animals. When we analyzed data from 8 experiments, we indeed found that *wntP- 2(RNAi);ptk7(RNAi)* animals formed 2-ectopic pharynges at a higher frequency than *ndl-3(RNAi);ptk7(RNAi)* animals (25% versus 6% of the time). However, such animals presenting two ectopic pharynges did not appear different and the low frequency of such events in *ndl-3(RNAi);ptk7(RNAi)* prevented a detailed histological comparison of these groups. For clarity we now provide scoring information for ectopic tissue formation as a single graph (Figure 2) and a separate measurement of pharynx angle (Figure 2), as well as images of animals with ectopic mouths and ectopic pharynges for each of the experimental treatments tested (Figure 2).

2) Figure 3 nicely shows that ptk7, WntP-2 and ndl-3 expression are affected by β-catenin RNAi. However, because the patterning of more posterior tissues appears to be controlled by WntP2, ndl-3 and ptk7, I think it is important to also show how expression of these genes are affected in posteriorized animals.

We tested *APC(RNAi)* regenerating trunk fragments and detected ectopic anterior expression of *wntP-2* and ectopic anterior expression of *ptk7* (Figure 5—figure supplement 1).

*3) The claim (in Figure 4) that ndl-3 expression is controlled by feedback is not well supported. Another possibility is that the pharynx position dictates the posterior boundary of ndl-3 expression, rather than feedback between Wnt signaling and FGFRL signaling.*

We performed experiments to identify the cell type that expresses ectopic *ndl-3* in *ptk7(RNAi);wntP-2(RNAi)* animals prior to formation of a fully-formed ectopic pharynx and find the majority are *collagen+* cells in the region of the body-wall musculature, a population of cells with hypothesized regulatory functions. Such cells are not known to actively contribute tissue to a regenerating pharynx, arguing against the ectopic *ndl-3* expression as marking the nascent ectopic pharynx itself. We removed the feedback inhibition model from the abstract and from the figures (Figure 7), and now discuss that model along with the possibility that the pharynx itself could control aspects of the *ndl-3* expression domain, as suggested.

*4) Do head or tail fragments eventually form ectopic mouths/pharynges? If the axial patterning is altered, I would expect to see this during regeneration too.*

Head and tail fragments undergoing *ptk7;wntP-2* RNAi do not initially regenerate ectopic mouths and pharynges (by 22 days of regeneration), and we now elaborate on those results in Figure 2. Eventually, such animals do form ectopic pharynges if they are fed dsRNA over a long timeframe (74 days, 3 of 12 animals examined), but we attribute this to the homeostatic functioning of the genes observed in uninjured animals (Figure 2). Our interpretation of these context specific functions for *ptk7/wntP-2/ndl-3* in suppression of trunk identity in regeneration is that these genes could signal context-specific information about the presence/absence of regions from the animal fragment relevant for reformation of the tail and we now elaborate on this interpretation in the Discussion.

5) The authors observe RNA "gradients" with their 3 main genes of interest, but what evidence do they have that this is a protein gradient?

Our analysis was restricted to detection of mRNA for these three genes and we edited the text to clarify this.

6) If ptk7 specifies "trunk" as the authors strongly state, then why no anterior expansion when trunk identity is removed? Brain should be assayed. Trunk expansion and not loss of trunk should give ectopic pharynxes, yet the authors report the opposite. Perhaps it is that ptk7 restricts FoxA+ stem cells, and when knocked down, that population expands and induces another pharynx.

We appreciate the identification of this wording issue. Our model is that *ptk7* negatively regulates trunk identity within the tail and we have modified the text to clarify the sign of this regulatory effect. The nature of this effect is directional, as we never observed formation of ectopic pharynges anterior to the original pharynx, which we suggest is due to the posterior regionalized co-expression of *ptk7* and *wntP-2.* We examined the brains of *ptk7(RNAi);wntP-2(RNAi)* animals by detecting *ndk* (Figure 3) and *chat* (Figure 3—figure supplement 1), and detecting *cintillo* expression that marks a subpopulation of lateral brain neurons whose numbers scale with the size of the brain (Figure 3—figure supplement 1). We did not detect any clear difference to the size or expression status of the brain by these criteria. We further examined the possible involvement of *FoxA*+ cells in the ectopic pharynx phenotype and could detect the expansion of the *FoxA*+ expression domain in fully regenerated *ptk7(RNAi);wntP-2(RNAi)* and also *ptk7(RNAi);ndl3(RNAi)* trunk fragments (Figure 2). We were able to detect a region of ectopic *FoxA+* expression at an early time in the appearance of the ectopic pharynx, consistent with a mechanism in which trunk duplication occurs via the production or relocalization of *FoxA+* precursor cells (Figure 3).

7) Dvl and fz1/2/7 are stated to also enhance the ptk7 phenotype, but were dropped from analysis. Why?

We focused the majority of our efforts in deciphering the functions for the regionally expressed signaling molecules (*wntP-2*, *ptk7,* and *ndl-3)* in order to account for the unidirectional nature of the regional expansion/duplication phenotypes, whereas both *fzd1/2/7* and *Dvl-2* are broadly expressed. We now include additional verification for the involvement of these factors in suppressing trunk identity similar to *wntP-2, ptk7* and *ndl-3* (Figure 5—figure supplement 2). We now also discuss a possible molecular pathway of action supported by these observations, that *wntP-2* signals through *ptk7* and *fzd1/2/7* and *Dvl-2*, but likely independently of *β-catenin-1*, to suppress trunk identity (Figure 7).

8) Almost no stains are quantified. RNAi efficacy is not measured by qRT-PCR or RNAseq as is standard in the planarian field. This would greatly help in making such statements as in “These results suggest that β-catenin upregulation can be sufficient for tail axis formation in conjunction with wntP-2 and ptk7 expression. Taken together, normal levels of β-catenin signaling are important for the normal expression of pkt7, wntP-2 and ndl-3”.

We now include scoring information for images presented throughout (see legends and/or text), quantification of pharynx orientation (Figure 2), quantification of trunk regional expansion and tail restriction (Figure 3), quantification of graded expression of *wntP-2, ptk7*, and *ndl-3* (Figure 4 and Figure 4—figure supplement 1), and cell counts for *ndl-3*+*collagen*+cells in the tail of *ptk7(RNAi);wntP-2(RNAi)* animals. We quantified efficiency of *wntP-2, ptk7,* and *ndl-3* RNAi by qPCR (Figure 2—figure supplement 1).

9). In vertebrates, ptk7 has been associated with WNT, semaphoring/plexin, and VEGF signaling, in addition to its known roles in PCP. None of this was tested here to bolster mechanism. Perhaps PCP defects can explain aspects of the phenotype. The authors state that the ectopic pharynxes are properly oriented, but many examples shown do not look that way.

We identify *ptk7* and *wntP-2* as suppressing pharynx and mouth formation. Given that Ptk7 proteins can act as Wnt-coreceptors, and given the rarity of ectopic trunk phenotypes (not observed in previously published large-scale RNAi screens such as Reddien et al. 2005), this suggests a molecular pathway of action in which *WntP-2* signals through Ptk7 to inhibit trunk fates. We support this conclusion by finding similar phenotypes from inhibition of a frizzled receptor, *fzd1/2/7* (Figure 5—figure supplement 2). These observations argue against a likely participation of Wnt-independent Ptk7 signaling (such as via semaphoring/plexin signaling or VEGF signaling) in trunk suppression. We tested several components of the Planar Cell Polarity pathway (*vangl-1, vangl-2*, *DAAM1,* and *ROCK*) because of its connection with noncanonical Wnt signalling pathways and could not find genetic interactions with *ptk7* (Figure 5—figure supplement 3). We also measured the orientation of the ectopic pharynx in all of our RNAi treatments (Figure 2) and found that the majority of such structures are indeed oriented obliquely. In some cases the pharynx was oriented toward the anterior, but the majority of ectopic pharynges in all conditions pointed toward the posterior (the arc from -90 to 0 to 90 degrees) rather than toward the anterior (the arc 90 to 180/-180 to -90 degrees), and the distributions of angles did not appear fully randomized (as compared to equal numbers of randomly selected angles plotted as in Figure 2, not shown). It is possible that a cell polarization pathway is involved in the regional identity determination controlled by these genes (and we now discuss this possibility in the Discussion paragraph two), but our data does not find evidence it is the PCP pathway. Alternatively, from the appearance of many of the ectopic pharynx phenotypes it seems that the structure grows in a confined manner that can result in the attainment of a particular angular orientation. Of note, animals with two ectopic pharynges always formed them oriented toward the posterior midline. We also note that the posterior and anterior pole identities in animals with ectopic pharynges were normal, suggesting that trunk suppression (and orientation of any ectopic pharynx) and body axis polarization are under distinct control mechanisms.

*10) More cell type markers are needed for tissues in different regions to conclude certain axial areas are not expanding/contracting.*

We now include additional histological analysis, which totals to examining each body region (anterior pole, head, prepharyngeal region, trunk, tail and posterior pole) by detecting expression of *notum*, *ndk, wnt2, mmp1*, *FoxA, wnt11-1, fzd4-1, Abd-Ba,* and *wnt1* for both *ptk7(RNAi);wntP-2(RNAi)* and *ptk7(RNAi);ndl-3(RNAi)* animals (Figure 3) and additionally *chat, cintillo, mag-1*, and *wnt11-2* and *ptk7(RNAi);wntP-2(RNAi)* animals. We also present two new markers SMU15014980and SMU15007112that label prepharyngeal and peripharyngeal cells that will be useful for future investigations of planarian axis transformations. We quantified the lengths for several of the above domains and found that *ptk7*+*wntP-2* RNAi and *ptk7*+*ndl-3* RNAi expands trunk tissue (expressing *mmp1*, *FoxA,* or SMU15007112) and restricts gene expression normally associated with the anterior tail tissue region (*wnt11-1* and *fzd4-1)*. The *Abd-Ba* staining also showed this trend but the staining was too faint for us to confidently assign a numerical length to this domain.

11) Overlap in expression of ptk7, wntp-2 and ndl-3 should be quantified. How is the overlap changing in different RNAi conditions? Why would these genes overlap in the cells that express them, should it be a binary expression state of muscle?

We now measure the overlap in expression of *ptk7,* wntP-2 and *ndl-3* by quantification of in situ hybridizations (Figure 4, Figure 4—figure supplement 1). Quantification of colorimetric in situ hybridizations confirmed the graded expression of these genes as maximal in prepharyngeal (*ndl-3*), trunk (*ptk7*) and tail (*wntP-2*) regions (Figure 4). We also quantified expression of all three genes on a per-cell basis using triple FISH (Figure 4). This broadly confirmed the domains of gene expression. We analyzed this data in two ways to support the existence of expression cell states of the three genes that populate the body axis. First, scatterplots of expression of pairwise combinations of the three genes show the existence of co-expressing cells (Figure 4). Second, we determined approximate distributions of particular expression states by applying thresholds for low- versus high-expression to bin each cells into one of 8 possible expression classes (Figure 4—figure supplement 1) and found that these populate the axis in distinct domains. These experiments argue for a complexity to the expression states of the genes. We tested for possible regulatory interactions among these genes in uninjured animals by colorimetric and fluorescence in situ hybridizations, as well as by qPCR (Figure 6—figure supplement 3). In these experiments, the only strong and reproducible expression change we observed was the expansion of *ndl-3* into the tail region of *ptk7(RNAi);wntP-2(RNAi)* animals, and we confirmed that this expression occurred in *collagen+* cells of the body wall musculature (Figure 6). The identification of more subtle effects on graded expression is an interesting topic and we suspect it will require the development of new tools to be applied to the planarian system such as single molecule FISH.

12) Are the ectopic pharynxes functional?

We imaged two-pharynxed animals exposed to calf liver paste and found evidence that both the pre-existing and ectopic pharynx could acquire food (Figure 2, Video 1).